# ZnO Synthesized Using Bipolar Electrochemistry: Structure and Activity

**DOI:** 10.3390/ma12030535

**Published:** 2019-02-11

**Authors:** Arya Hakimian, Steven McWilliams, Anna Ignaszak

**Affiliations:** Department of Chemistry, University of New Brunswick, Fredericton, NB E3B 5A3, Canada; arya.hakimian@unb.ca (A.H.); steven.mcwilliams@unb.ca (S.M.)

**Keywords:** bipolar electrochemistry, oxygen deficiency, zinc oxide, blue-shift, photocurrent, charge recombination, electron lifetime

## Abstract

The photoactive materials broadly applied in catalysis and energy conversion are generally composed of metal oxides. Among these oxides, ZnO showed a promising photocatalytic activity; however, traditional synthetic routes generated by-products and large amounts of secondary waste. Herein, we report the use of bipolar electrochemistry to generate ZnO nanoparticles using deionized water and a zinc metal to conform to green chemistry practices. TEM imaging demonstrated that the sizes of the bipolar-made ZnO particles were smaller than the commercial sample. The presence of structural defects in ZnO was correlated with the chemical shifts analyzed by X-ray photoelectron spectroscopy (XPS) and by different concentrations of O^2−^ ions in stoichiometric and defected lattice. Further, the diffuse reflectance UV–Vis studies demonstrated a blue-shift in the reflectance spectrum for the bipolar-made oxide. This was also an indication of defects in the ZnO lattice, which related to the formation of shallow levels in the bandgap of the material. The structural and morphological differences influenced the photocatalytic characteristics, revealing a higher photocurrent for the bipolar-made ZnO when compared to the reference sample. This was further manifested in lower total resistivity for all anodes made from the non-stoichiometric ZnO, and also in their shorter diffusion length for charge exchange and electron lifetimes.

## 1. Introduction

Metal oxides are main components of inorganic photovoltaics (IPVs) and photocatalysts employed in a light-driven water splitting, waste water disinfection or oxidation of organic matter. They are classified as broad bandgap semiconductors and generally have large excitation binding energies (BE) [1]. The synthetic pathways for their fabrication on an industrial scale are categorized as physical (i.e., chemical vapor deposition and high energy ball milling), and chemical, including precipitation, sol–gel processing and hydrothermal methods [2]. A metal oxide that has gained attention is ZnO in part due to its wide bandgap in the range of 3.37–3.44 eV [3].

Although ZnO (and metal oxides in general) have shown excellent photocatalytic and photovoltaic properties, typical synthetic routes can incur large amounts of secondary waste from the use of excessive solvents, generation of by-products, including halogenated species, and in some cases require extreme temperatures [2]. In order to address this issue, the use of green chemistry is prioritized over conventional synthesis, regardless the cost [4]. One of the most environmentally friendly methods is electrosynthesis, allowing to reduce or even to exclude reactants. ZnO has been recently synthesized in a bipolar electrochemical reactor using only water and Zn metal, reacting with each other in a constant electric field [5].

In bipolar electrochemistry (Figure 1A), the wireless bipolar electrode (*BPE*) was generated by the potential difference across two driving electrodes, which created a gradient in the solution potential. This contrasted with typical electrochemistry where all electrodes were in direct electrical contact with the power supply [6]. This applied potential resulted in a gradient of electric field through the solution and the surface of conducting material, acting as the wireless bipolar electrode. At a certain critical potential, electrochemically driven redox reactions began to occur at the extremities of the BPE [7]. The total current flow (*i_tot_*) generated from these reactions can be split into two paths (Figure 1B).

Part of the current passed through the bipolar electrode (*i_bpe_*) and the remaining current flowed through the solution (*i_sol_*). In the absence of the BPE, total current passing between the driving electrodes was entirely ionic and the magnitude of this current was governed by the applied potential (*E_tot_*) and the resistance of the solution (*R_s_*). When the BPE was added, a new path for current flow was available. Total system resistance was now broken up into four components *R_s1_*, *R_s2_*, *R_s4_* and *R_bpe_*. If *R_s2_* < *R_bpe_*, then the majority of current would pass through the solution, and therefore the BPE did not significantly affect the electric field. However, when *R_s2_* > *R_bpe_*, substantial current flowed through the BPE. This could cause differences in local electric field strength, thus resulting in nonlinear polarization in solution above the BPE. A general formula for determination of the fraction of *E_tot_* dropped over the BPE referred to as Δ*E_bpe_* can be estimated based on the following Equation (1) [7].
(1)$$\Delta E_{bpe}=E_{tot}\left(\frac{I_{bpe}}{d_{channel}}\right)$$
where *l_bpe_* is the length of the BPE and *d_channel_* is the distance between the driving electrodes. An important remark was that the bipolar electrode did not accumulate any net charge [7,8], which meant that the rate of redox reactions must be equal, and therefore the pH of the surrounding solution remained constant.

This research was focused on the electrochemical bipolar synthesis of nanostructured ZnO that will be carried out at a much lower applied potential as compared to a previous study [5]. This was accomplished by changing the distance between the driving electrodes and adjusting the thickness of bipolar electrode (metallic zinc). Changing the time of polarization was expected to influence the particle size and shape of the product. Furthermore, the oxide was examined in terms of morphology (TEM and SEM), electronic structure (X-ray photoelectron spectroscopy, XPS), optical properties (diffusive reflectance UV–Vis spectrophotometry, DR-UV–Vis) and photoelectrochemical activity under dark and illuminated conditions using voltammetry, chronocoulometry and impedance spectroscopy.

## 2. Materials and Methods

Zinc metal plates (99.99% trace metals basis) acting as the wireless BPE, ruthenium (II) N-719 sensitizing dye (95%), tertbutyl ammonium, lithium iodide, iodine, 4-tert-butyl-pyridine, acetonitrile (99.5%), hexamethylenetetramine (HMT, >99.0%) and polyvinylpyrrolidone (PVP, M_w_ = 40,000) were obtained from Sigma Aldrich. Fluorine doped tin oxide (FTO) glass with a surface resistivity of 7 Ω was purchased from Sigma Aldrich and cut into 2.5 cm × 5.5 cm × 0.2 cm slides. Ethanol (100%) and isopropanol (70%) were supplied by Commercial Alcohols (Brampton, ON, Canada), and acetone (99.6%) was obtained from Fisher Scientific. Ultra-pure distilled water (18 MΩ) was used as the reaction solvent. The commercial ZnO reference was purchased from Baker Chemicals (Canada).

Scanning electron microscope (SEM) observations were performed using a JEOL JSM-6400 operating at an accelerating voltage of 15 kV. Transmission electron microscopy (TEM) was done using a JOEL JEM-2010 microscope fitted with a Gatan Ultrascan camera using an accelerating voltage of 200 kV. Energy dispersive X-ray (EDX) analysis was performed in parallel with SEM and TEM using an EDAX Genesis 4000 EDX Analyzer at an accelerating voltage of 15 kV. Platinum counter electrode (CE) was prepared using an Edwards S150A Sputter Coater equipped with a platinum head. X-ray diffraction (XRD) patterns were measured using a Bruker D8 Advanced Diffractometer. The X-ray source was a 2.2 kW Cu X-ray tube, maintained at an operating current of 40 kV and 25 mA. Data were fitted using EVA software (Bruker). X-ray photoelectron spectroscopy (XPS) was performed at Dalhousie University on a VG Microtech MultiLab ESCA 2000 system with a 100 µM analyzer spatial resolution, 0.01 eV energy resolution and Mg Kα linewidth. All data were calibrated with respect to C 1s excitation (284.6 eV) to account for charging effect. UV–Vis DRS was carried out at University of Toronto’s Analytical Laboratory for Environmental Science Research and Training Facility on a Lambda 25 UV/Vis Spectrometer double beam (DB) instrument in the range of 200–800 nm. All electrochemical analyses were done under visible light irradiation (50 W/cm^2^) and were performed using a CHI Instruments potentiostat model number CHI660E. Open circuit potential (OCP) measurements were examined over 600 s with a sample interval of 0.1 s. Single-potential amperometry was used to generate the current–time (*I–t*) curves and were acquired over 900 s with switching between dark and illuminated states every 60 s. Linear sweep voltammetry (LSV) was performed from −1.0 V to 0.8 V at a scan rate of 0.01 V/s, and the electrochemical impedance spectroscopy (EIS) in the frequency range from 0.01 to 10,000 Hz at an open circuit voltage. ZView software (Scribner Associates) was used for the modelling of impedance spectra.

### 2.1. Synthesis of ZnO Using Bipolar Electrochemistry

The bipolar electrochemical synthesis of ZnO was carried out using 50 mL of ultra-purified water and a Zn-metal plate in a manner similar to that outlined in Figure 1A and Appendix A. The BPE was suspended between two 2.0 cm × 12.0 cm × 1.0 cm stainless steel plates acting as driving electrodes (DE), separated by a distance of 1.25 cm. Both the BPE and DEs were held parallel to each other using a Teflon cap and plexiglass blocks as space holders. A Keysight E36106A DC power supply was used to apply 62.5 V_dc_ to the system for varying time (*t*), temperature (20 and 5 °C) and type and concentration of surfactant (PVP and HMT). Under the applied potential, hydrolysis of water occurred on both the driving electrodes as well as the BPE. The presence of hydrolysis was noted by the onset of bubble formation (oxygen and hydrogen gas). As the surface of the anodic pole of BPE electro-dissolved, the Zn^2+^ ions released to solution underwent complexation with the free hydroxyl ions through a series of formation reactions, and ZnO precipitated as an off-white solid based on the following reactions.
Zn^2+^_(aq)_ + OH^−^_(aq)_ → Zn (OH)^+^_(aq)_(2)
Zn (OH)^+^_(aq)_ + OH^−^_(aq)_ → Zn (OH)_2(s)_(3)
Zn (OH)_2(s)_ → Zn (OH)_2(aq)_(4)
Zn (OH)_2(aq)_ + OH^−^_(aq)_ → [Zn (OH)_3_]^−^_(aq)_(5)
[Zn (OH)_3_]^−^_(aq)_ + OH^−^_(aq)_ → [Zn(OH)_4_]^2−^_(aq)_(6)
Zn (OH)_2(aq)_ → ZnO_(s)_ + H_2_O_(l)_(7)
Zn (OH)_2(s)_ → ZnO_(s)_ + H_2_O_(l)_(8)
[Zn (OH)_3_]^−^_(aq)_ → ZnO_(s)_ + H_2_O_(l)_ + OH^−^_(aq)_(9)
[Zn (OH)_4_]^2−^_(aq)_ → ZnO_(s)_ + H_2_O_(l)_ + 2OH^−^_(aq)_(10)

Following the reaction, the aqueous medium was removed by evaporation at ~95 °C until no liquid was present, or via filtration using 0.22 µM pore size Millipore filters. For filtration, the collected material was then dried in the oven at ~95 °C for 12 h. In both cases, once dried, the powder was collected and stored in anhydrous ethanol. For the scope of this study, a working voltage of 62.5 V_dc_ was applied for all trials, and this overpotential was established experimentally based on the yield of oxide. The photos shown in Appendix A indicate that the hydrolysis of water occurred at overpotential as low as 10 V (Appendix A). Further increase of an applied electric field led to a significant increase in yield, resulting in 62.5 V as optimal operating voltage. This was further correlated with amperometry tests (Appendix A), where acquired current was measured between driving electrodes. As demonstrated in Appendix A, the resulting current recorded at a polarization of 10 V was almost one order of magnitude lower than the current generated at 62.5 V. This increase in current passing through the solution is related to higher concentration of ions (Zn^2+^, OH^−^ and H^+^), resulting in the increase of total conductivity of the solution, and thus in the generation of sufficient ZnO product.

### 2.2. Fabrication of Photoelectrode

For the preparation of the working electrode, ZnO was mixed with anhydrous ethanol giving a concentration of 1 mg/40 µL. Prior to deposition, the mixture was sonicated for approximately 5 min and the resulting suspension was drop-cast onto cleaned FTO until an approximate loading of 1.0 mg per area unit was obtained. The thin films were annealed at 300 °C for 10 min with an initial ramp of 10 °C/min, a 10-min hold, and a cooling rate of 10 °C/min.

## 3. Results and Discussion

### 3.1. Morphology: Effect of Reaction Time and Surfactant

TEM and SEM observations of the synthesized oxide indicated significant differences in morphology depending on synthesis time (Figure 2). The commercial ZnO reference was rod-shaped with sizes in the range of 0.5 to 1 µM in length (Figure 2A). The TEM image of ZnO generated after 2 h of polarization at ~5 °C (sample labeled as 2H, Figure 2B insert) and collected via filtration was composed of connected spheres of varying sizes (20–40 nm) arranged in a chaplet-like manner. Once deposited onto the FTO substrate and annealed, these structures formed porous clusters as shown in SEM (Figure 2B). ZnO generated at room temperature over 3 h of polarization (sample named 3H) and collected via evaporation resulted in larger star-shaped structures of approximately 0.5 µm in length (Figure 2D). As demonstrated in the supporting information (Appendix A), an increase in particle size was related to the agglomeration of the ZnO nanoparticles from the prolonged exposure to water during the evaporation process [9], not to longer synthesis time. This was further supported by the TEM image of 3H collected via filtration carried out immediately after synthesis, revealing a mixture of previously seen spherical nanoparticle clusters as well as larger agglomerates (Figure 2C). The microscopy led to the conclusions that exposure of ZnO to water can drastically affect the resulting structures, and ZnO synthesis in aqueous media must be carefully monitored. The structure evolution from the spherical nano-beads to the star-shaped larger structures was demonstrated by another group [10]. The proposed mechanism indicated that upon the prolonged exposure to water, the structure of ZnO changed and an initial amount of water affected the nucleation process of ZnO significantly. As observed, an extended contact with water can impede the (0001) growth and accelerate the (1100) growth. This can be controlled by changing the water content via mixing it with methanol [10]. In this way, the shape and size of ZnO can be tailored by adjusting the volume ratio. The optimized conditions resulting in uniform size and shape of nanomaterial (connected nano-spheres) were 2 h runs carried out at 5 °C (Figure 2B). Appendix A demonstrates TEM images of these structures synthesized at room temperature (Appendix A) and in an ice bath (~5 °C, Appendix A). The smaller particles size of product fabricated at ~5 °C can be related to slower formation of the ionic species (kinetically controlled electrolysis). This was further confirmed by a smaller current recorded for low-temperature trials (Appendix A), and thus resulting in a slower rate of agglomeration of ZnO nanoparticles.

The use of surfactant (HMT and PVP) at various concentrations was also investigated and optimized for 2- and 3-h synthesis trials and collection method (filtration and evaporation). The generated current profiles were not significantly influenced by the addition of surfactants (Appendix A); therefore, they showed promise as an easy way to reduce the overall agglomeration without lowering the yield. It was later discovered that the effect of filtration surpassed the benefit gained from surfactant addition. As such, the use of surfactants, specifically HMT, although beneficial to an extent (Appendix A A shows good particle separation in the presence of HMT for synthesis carried out for 2 h), was inhibited by an additional washing step (ZnO agglomerated as soon as the surfactant was removed). The PVP showed no significant effect of particle separation as compared to HMT (Appendix A).

### 3.2. Structure

X-Ray diffraction (XRD) allowed the identification of ZnO crystallographic structure as well as the estimation of crystalline size. The XRD patterns are shown in Figure 3, and, for both the synthesized and commercial ZnO, the diffraction peaks indicated the wurtzite hexagonal phase (JCPDS 00-036-1451). The crystallite size (*D*) was calculated using the Debye–Sherrer formula [11].
(11)$$D=\frac{0.9\lambda }{\beta \cos\theta }$$
where *D* is the crystallite size in nm, *λ* is the wavelength of X-ray radiation (0.0154178 nm), *θ* is the Bragg angle in radians and *β* is the full-width at half-maximum (FWHM), corresponding to the most intense peak in radians (summary of XRD analysis can be seen in Table 1). The largest crystallite size was obtained for the commercial standard, followed by the 3H evaporated sample. The difference in particle size of 2H and 3H filtrated samples was very similar, since their morphologies were fairly similar. The XRD analysis of crystalline size followed a similar trend in ZnO particle size analyzed by TEM imaging (shown in the insert in Figure 2). In comparison to ZnO synthesized by the bipolar electrochemical process by Allagui et al. [5], there was a similarity in XRD patterns with respect to the 3H evaporated trials, with <011> as a predominant face. For the 2H and 3H filtered samples, the XRD analysis revealed the <101> face as the strongest pattern. In fact, XRD scan for ZnO synthesized for 3H and collected by evaporation was identical as in the referred work (star-shaped [5]). This demonstrated the preferential growth of ZnO crystals along <011> plane upon prolonged exposure to water.

X-ray photoelectron spectroscopy (XPS) is further employed to analyze the chemical shifts that are summarized in Table 1. All binding energies were corrected for charge shifting using the most intense peak of C 1s to have a true value of 284.8 eV [12]. The binding energies of the Zn 2p signals for the synthesized samples were slightly shifted towards higher energies compared to the commercial (Figure 4A). These shifts can be related to the difference in morphology and particle size between samples; however, relative shifts in the peak positions had been primarily associated with oxygen deficiencies in the lattice [12,13]. This was also related to the shift of O 1s signal. To further investigate any oxygen-dependent defects in the synthesized samples, the O 1s spectra were resolved by deconvolution into two peaks (Figure 4C–F). The peak at the lower binding energy (P1) was related to the O^2−^ ions in the normal ZnO lattice, and the peak at higher energies (P2) to the O^2−^ ions in the oxygen-deficient regions within the ZnO (defective) structure [12,14]. The ratio between these two peaks had been shown to give an insight into the amount of oxygen defects [15]. The ratio of P1 to P2 decreased when compared to commercial ZnO (Com), Com > 3He > 3Hf > 2H, indicating increasing levels of defects in the ZnO structure [13].

### 3.3. Optical Analysis

Optical characterization was analyzed using UV–Vis DR spectrophotometry and used to measure bandgap and quantum confinement effects. The diffuse reflectance, *R*, of the samples is related to the Kubelka–Munk function *F(R)* by the relation *F(R) = (1−R)^2^/2R* (Figure 5A), where *R* is the percentage reflectance [16]. The bandgap energies of all samples were calculated from their diffuse reflectance spectra (Figure 5B) by plotting the square of the Kubelka–Munk function *F(R)^2^* versus energy in electron volts (Figure 5A). The linear part of the curve was extrapolated to *F(R)^2^ = 0* to get the direct bandgap energy. It is observed that the energy increased as the particle size decreased (Figure 5A). The commercial ZnO particles were bigger with much larger particle-size distribution and demonstrated a lower bandgap when compared to 2H and 3Hf. As demonstrated by others, this phenomenon corresponded to a quantum effect [17]. In particular, if the size of the particle became comparable with exciton radius (nanosize and quantum dots), optical properties of semiconductors changed significantly. The main reason mainly came from the carriers being confined in a very small space, which made the electrons and holes move only in a potential well. At the same time, it can improve the coupling between charges. Since the particle size of ZnO made using a bipolar electrochemical route were significantly larger than the exciton Bohr radius of bulk ZnO (2.34 nm [18]), the confinement effect was moderate.

For the synthesized oxide, the absorption edge is shifted to a slightly shorter wavelength (blue-shift; Figure 5B). The presence of blue-shifting can also indicate the defects in the ZnO lattice, resulting in the formation of shallow levels in the bandgap in this material [15,18]. This is consistent with an increase in bandgap energy for oxides synthesized in this work (Figure 5A). These defects cause the off-white color of the synthesized ZnO (Appendix A). Specifically, yellow ZnO has been related to oxygen deficiencies in the ZnO lattice, resulting in a non-stoichiometric phase Zn_1+x_O [19].

Nevertheless, the bandgap of commercial and synthesized ZnO falls within the range reported in literature ~3.3 eV at room temperature. Advantages associated with a large bandgap include higher breakdown voltages, ability to sustain large electric fields, lower electronic noise, and high-temperature and high-power operation. This is further analyzed by electrochemical methods.

### 3.4. Electrochemistry

Single-potential amperometry and linear sweep voltammetry data were converted to I-time curves (Figure 6A) and Tafel plots (Figure 6B), respectively. Under illumination, the 2H sample resulted in the highest photocurrent followed by the 3Hf, 3He, and the lowest for commercial ZnO (Figure 6A). The trend in measured photocurrent correlated well with the Tafel plots (Figure 6B) and values of kinetic parameters, such as exchange current density (*J_o_*) and charge-transfer resistance (*R_ct_*). The exchange current density can be estimated from the extrapolated intercepts of the anodic and cathodic branches and is related to the charge-transfer resistance according to [20].
(12)$$J_{o}=\frac{{RT}_{a}}{{nFR}_{ct}}$$
where *R* is the gas constant, *T_a_* is the absolute temperature, *n* is the number of electrons involved, *F* is the Faraday constant, and *R_ct_* is the charge-transfer resistance. As demonstrated in Figure 6B, slopes of the anodic branches are in the order of 2H > 3Hf > 3He > Commercial. Correlating this with the above equation (Equation (12), with the J_o_ being inversely proportional to *R_ct_*), it is concluded that the 2H samples resulted in lowest resistance, followed by 3Hf, 3He and Com. This analysis is in good agreement with the trend in photoresponse observed in Figure 6A.

Concerning conductivity, stoichiometric ZnO is an insulator with very high resistivity, while the presence of different defects changes the concentration of charge carrier, resulting in the formation of a degenerate semiconductor. Reduced ZnO, which is highly deficient in oxygen, is thus expected to be more conductive as compared to commercial ZnO. This explains the higher current measured for ZnO made with the electrochemical bipolar process. This is further correlated with resistances of the ZnO-based electrodes analyzed by the impedance spectroscopy.

The theoretical Nyquist plots of an illuminated ZnO photoanode (Figure 7A) shows two semicircles. A bigger one, in the frequency range of 1–10^3^ Hz, is attributed to electron transfer at the ZnO/dye/electrolyte interface and within the ZnO film (*R_ct_*); and to the electron ejection at the counter electrode/electrolyte interface. The transport in the electrolyte (transport resistance, *R_tr_*) is noted by the smaller semicircle in the frequency range of 10^3^–10^5^ Hz. The onset of the first semicircles at the real part of impedance are non-zero and are attributed to the charge transport through the FTO back contact together with solution resistance of the electrolyte (*R_s_*). The impedance measurements were conducted under dark and illuminated conditions, and the OCP potential was applied for each electrode based on Figure 7A, to probe the recombination between the electrons at the conduction band (CB) or the electrons with the acceptor species in the electrolyte. The diameter of charge-transfer resistance for the illuminated electrode was several orders of magnitude lower as compared to the test in dark (Figure 7C). This was further correlated with the Bode diagram (Figure 7B), where the absolute value of total resistance of the system was plotted against the applied frequency. The illuminated system showed significantly lower total resistivity, which we ascribed to an increase in current generated during illumination.

Table 2 shows calculated values of the electrical elements proposed in Figure 7, along with impedance analysis data from Figure 7 and Figure 8. *R_ct_* decreased upon irradiation for all ZnO-based electrodes, because of the improved charge-transfer along the ZnO films. The higher charge-transfer resistance (*R_ct_*) in Com and 3He samples as compared with 2H and 3Hf suggested a slower interfacial recombination, which can be attributed to a weaker interaction between the ZnO surface and the electrolyte solution. The porous structure in the 2H and 3Hf may facilitate penetration of sensitizer molecules into the structures. Consequently, the injected electrons and the oxidized ions in the electrolyte can be suppressed. Both the morphology and structural effect (oxygen vacancies) contribute to the observed resistivity. For example, as shown by SEM imaging, the electrode prepared from the 3Hf ZnO (Figure 2C) was still relatively porous after annealing, although there were ZnO agglomerates which were bigger as compared to the 2H sample (Figure 2B). The overall morphology of this electrode facilitated the efficient charge and electrolyte transport, which was manifested in lower *R_ct_* as compared to 3He and Com samples. The SEM imaging for the latter two (Figure 2A,D) showed significantly less porosity, which led to a reduced interaction between ZnO surface and the electrolyte solution. Yet, the oxygen-deficient ZnO was expected to be more conductive due to the increased mobility of charge carriers. This manifested in lower *R_ct_* and *R_tr_* resistivity for all anodes made from ZnO synthesized in this work (Table 2).

Another important parameter is the electron diffusion length (*L_n_*) analyzed from the impedance response [21]. This parameter characterizes the maximum travelling distance of electrons before recombining with an acceptor species in the electrolyte. An efficient photoelectrode ideally reveals diffusion lengths, which are larger than the thickness of the photosensitive oxide layer, (*L*). Notice, in order to examine the thickness of the active materials, an epoxy cross-section of the photoanodes were measured using SEM imaging as shown in the supporting information (Appendix A). All sample thicknesses were determined using ImageJ software and corrected to a true loading value of 1.0 mg per sample.

The electron diffusion length can be expressed as follows [21].
(13)$$L_{n}=L\sqrt{\frac{\tau_{n}}{\tau_{d}}}=L\sqrt{\frac{R_{ct}}{R_{tr}}}$$
where *R_ct_* is the charge recombination resistance, *R_tr_* is the macroscopic transport resistance, *τ_n_* is the charge lifetime, and (*τ_d_*) is the charge extraction time. The diffusion length would exceed the thickness of the ZnO electrode if *R_tr_* < *R_ct_* or *τ_d_* < *τ_n_*. It was apparent in the Nyquist plot that *R_ct_* was larger than *R_tr_* indicating an efficient charge collection. The shortest *L_n_* was observed for 2H, followed by commercial sample and 3Hf, and the longest for 3He. This also confirmed that the porosity of the electrode had significant impact on the efficiency of charge transport and was correlated with lower *R_ct_* for 2H ZnO, and thus better coverage of the electrode surface. This related to its high photocurrent generation as compared to other samples (Figure 6A) and lower resistivity (Figure 8).

Furthermore, the conduction band free-electron lifetimes (*τ_n_*) in the photoelectrodes can be evaluated from the peak frequency (*f_peak_*) at the minimum phase angle in the Bode plots (Appendix A in supporting information) for illuminated anodes, using the correlation [21]
(14)$$\tau_{n}=\frac{1}{\omega_{\mathrm{peak}}}=\frac{1}{2{\pi f}_{\mathrm{peak}}}$$

The calculated electron lifetimes are summarized in Table 2. Although the differences in charge lifetime were very small, it can be observed that the shortest lifetime of charge carriers, and thus relatively higher charge recombination rates, occurred for the ZnO anode made from the 2H sample.

## 4. Conclusions

In summary, ZnO was successfully synthesized by bipolar electrochemistry using only water and metallic zinc. This method was operationally simple and environmentally benign. The most important factor, influencing the resulting structures, was the time when ZnO nanoparticles were in contact with water (not the applied potential in bipolar reactor as anticipated). The surfactants showed no positive effect on the particle size and shape at any synthesis conditions. Also, the particle sizes of the bipolar-made ZnO were smaller than the commercial oxide, and the particle shape of ZnO made at optimized bipolar conditions was more uniform.

Also, a more defective (non-stoichiometric) structure was observed for oxides fabricated by bipolar electrochemistry. The presence of structural defects was correlated with the chemical shifts in the Zn 2p and O 1s XPS signals. This was further correlated with the changes in the concentration of O^2−^ ions in stoichiometric and defected lattice.

The diffusive reflectance UV–Vis studies revealed a blue-shift in the reflectance spectrum for the bipolar-made oxide that corresponded to the increased absorption of longer wavelength light. This confirmed the presence of oxygen defects in the ZnO lattice. On the other hand, the Kubelka–Munk function revealed higher bandgap energy for ZnO made by the bipolar electrochemistry with the smallest particle size (3.42 eV, 20–40 nm particles) when compared to the commercial ZnO (3.27 eV, µM range). This phenomenon was more strongly related to the morphology (particle size in particular) than to the defective structure of ZnO and was reported as the quantum effect.

The structural and morphological differences influenced the photocurrent characteristics, revealing higher photocurrent acquired for the bipolar-made ZnO when compared to the reference sample. The higher charge-transfer resistance (*R_ct_*) in commercial ZnO suggested a slower interfacial charge recombination, which can be attributed to a weaker interaction between the ZnO surface and the electrolyte solution. The oxygen-deficient ZnO showed higher conductivity due to the increased mobility of charge carriers. This manifested as lower total resistivity for all anodes made from the oxygen-deficient ZnO, as well as in their shorter diffusion length for the charge travelled and electron lifetimes.

In summary, this study offered a reagent-free synthesis of nanostructured ZnO with improved photoelectrochemical activity that has the potential to be utilized in a large-scale production.

## Figures and Tables

**Figure 1 materials-12-00535-f001:**
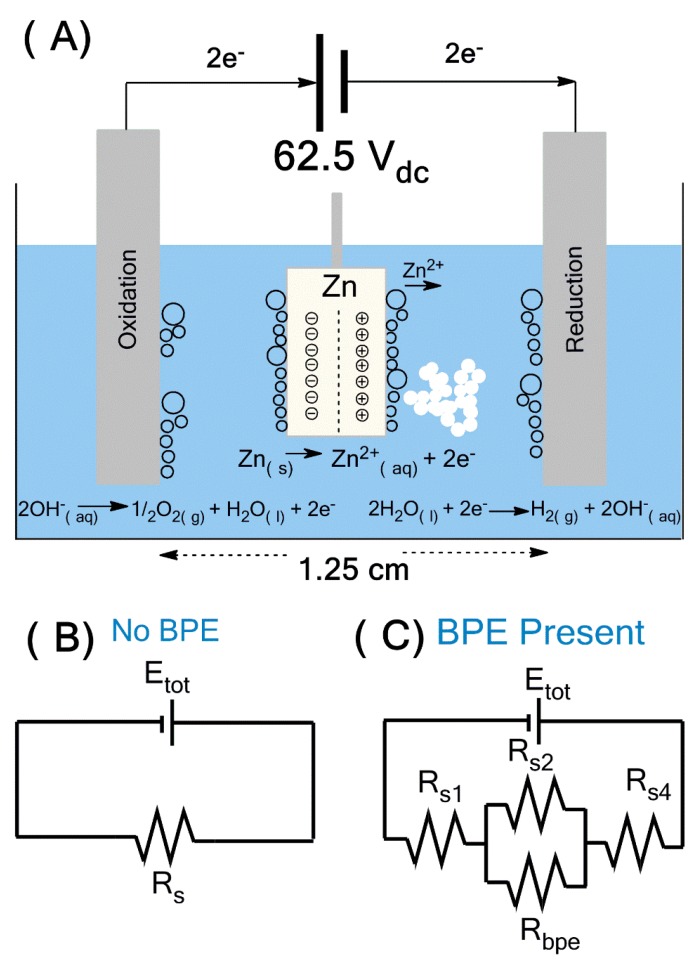
Electrochemical bipolar setup for the synthesis of ZnO (**A**), and an electrical equivalent circuit representing resistances to current flow in a bipolar electrochemical cell without (**B**) and with bipolar electrode (**C**).

**Figure 2 materials-12-00535-f002:**
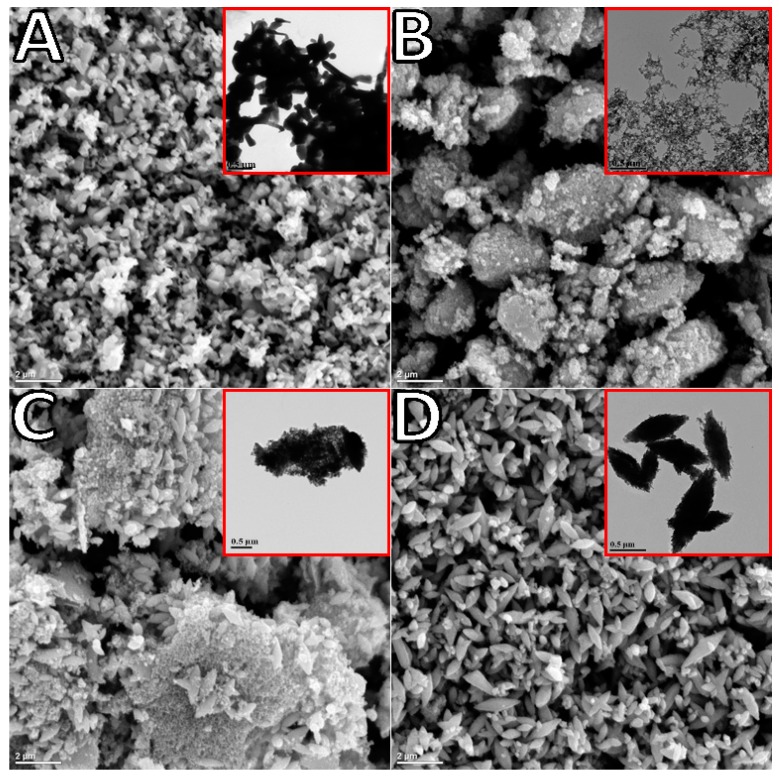
SEM of ZnO annealed (300 °C, 10 min) and TEM (inserts) of the commercial ZnO (**A**), ZnO filtrated after 2 h of synthesis, 2H filtrated (**B**), ZnO filtrated after 3 h of synthesis, 3H filtrated (**C**) and ZnO collected from aqueous suspension after water was evaporated, 3H evaporated (**D**), synthesized by the bipolar electrochemical reaction.

**Figure 3 materials-12-00535-f003:**
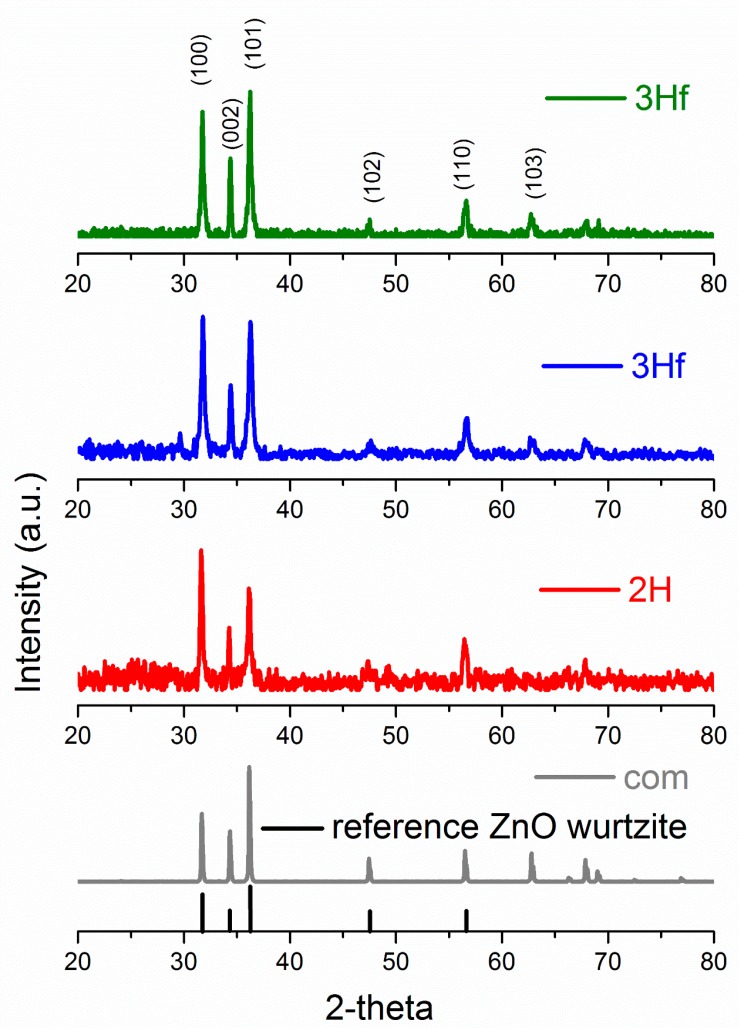
X-ray diffraction patterns of commercial ZnO (gray), and ZnO synthesized by bipolar electrochemical process after 2 h and collected by filtration (2H, red), 3 h and collected by filtration (3Hf, blue) and 3 h and collected by evaporation (3He, green). The reference signals of ZnO according to JCPDS card number 36-1451 are marked at the bottom in black.

**Figure 4 materials-12-00535-f004:**
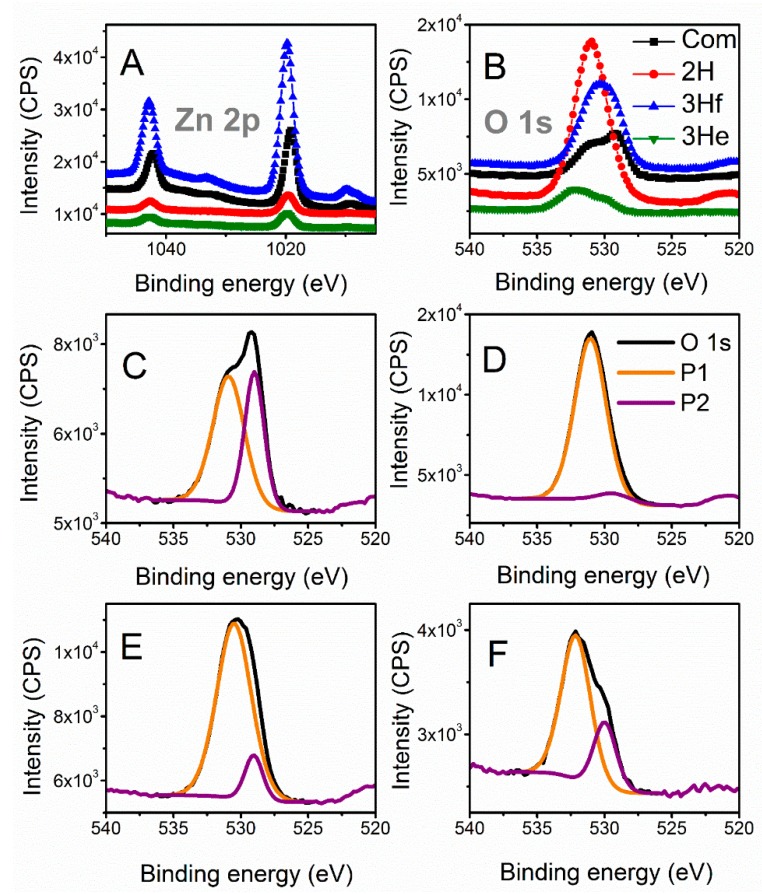
X-ray photoelectron spectroscopy of Zn 2p (**A**) and O 1s (**B**) narrow scans; deconvolution of O 1s signals for commercial ZnO (**C**), 2H trials (**D**), 3H filtered (**E**) and 3H evaporated sample (**F**). O 1s (C–F) signals were resolved into two peaks (P1 and P2) related to O^2−^ ions in the normal ZnO lattice and to the O^2−^ ions in the oxygen-deficient regions, respectively.

**Figure 5 materials-12-00535-f005:**
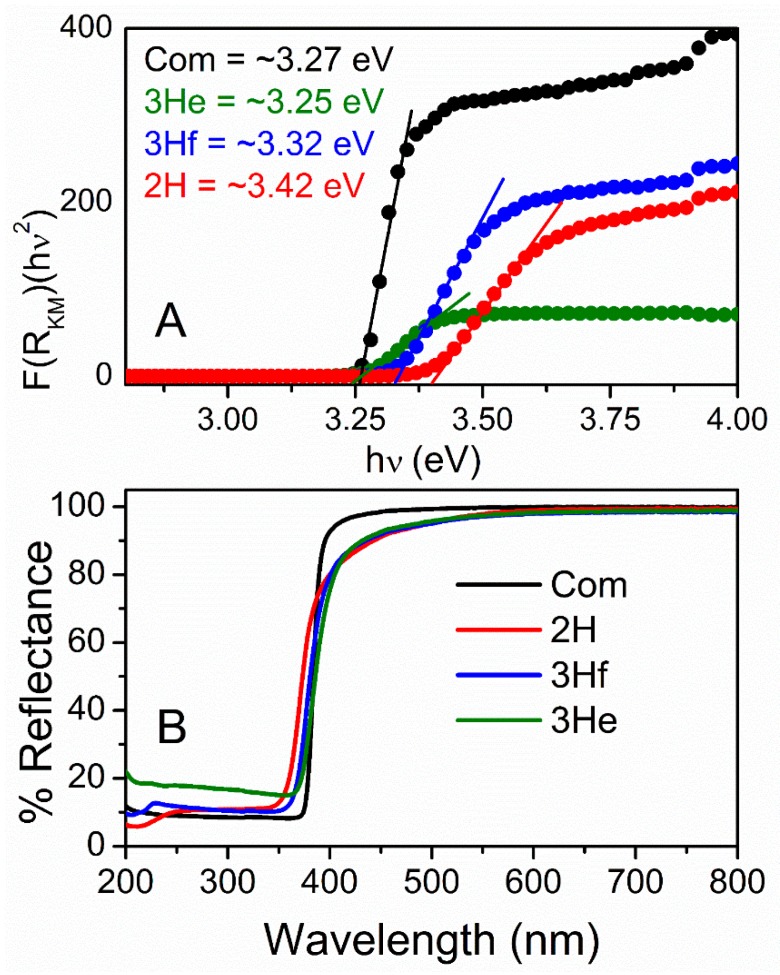
Kubelka–Munk function versus bandgap energy (**A**) calculated from UV–Vis diffusive reflectance spectra (**B**) for commercial ZnO (black) and synthesized by bipolar electrochemistry (2H—red, 3H filtrated blue and 3H evaporated—green).

**Figure 6 materials-12-00535-f006:**
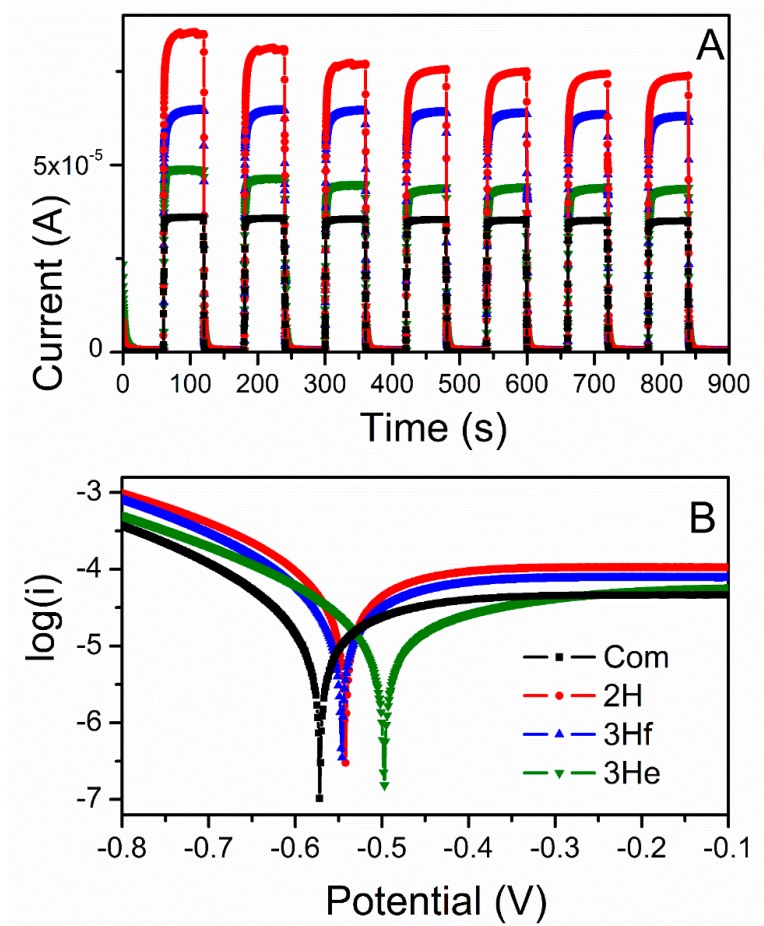
Photocurrent response under dark and illuminated conditions recorded at open circuit potential (**A**), and Tafel plots generated from linear sweep voltammograms under illumination (**B**).

**Figure 7 materials-12-00535-f007:**
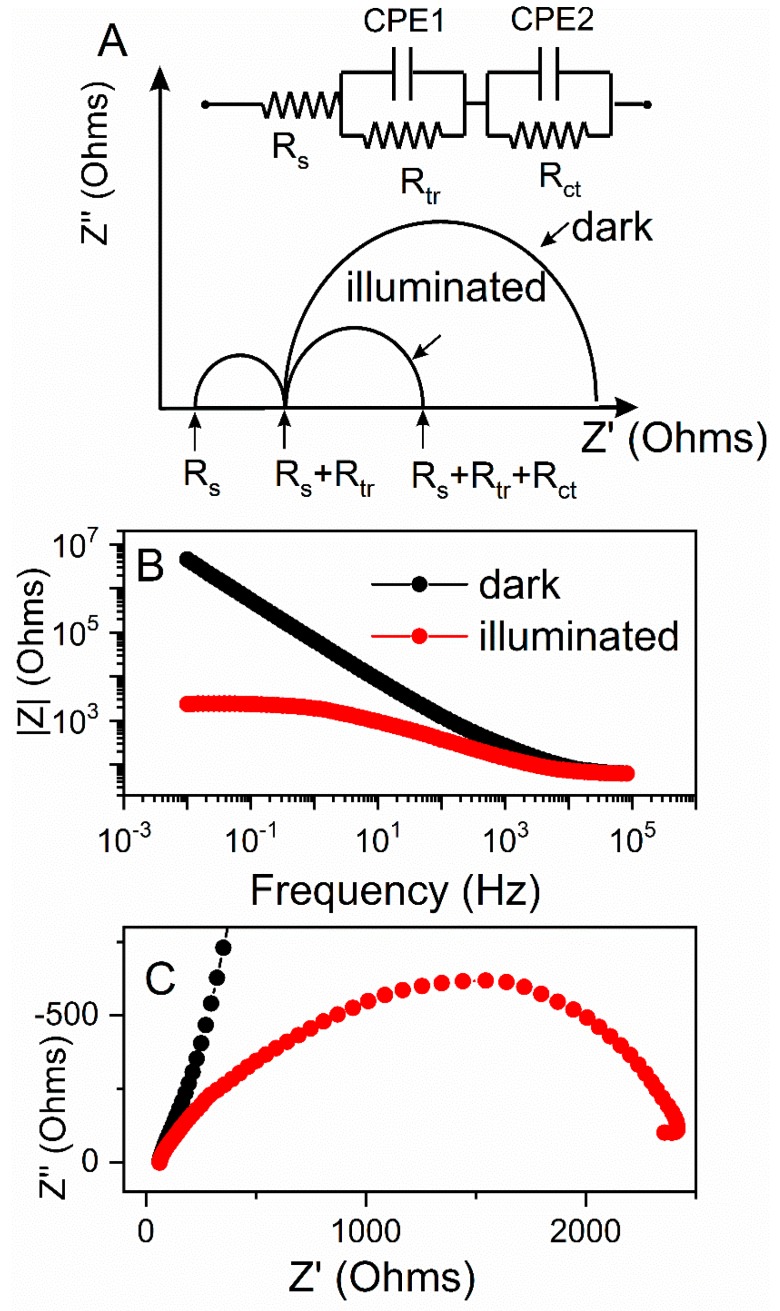
Theoretical impedance spectra and an electrical equivalent circuit representing distribution of the resistance in photoelectrochemical cell under dark and illumination (**A**); an example of Bode diagram |*Z*| = f(*frequency*) (**B**), and Nyquist plot *Z′* = f(*Z″*) (**C**) under dark (black) and illuminated (red) conditions.

**Figure 8 materials-12-00535-f008:**
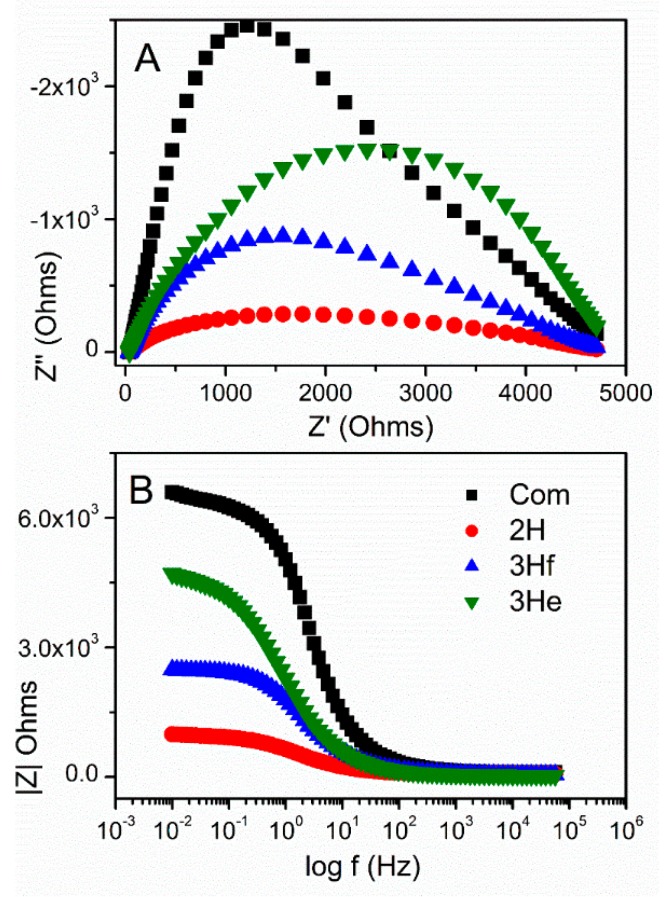
Nyquist plots (**A**) and Bode diagrams (**B**) under illumination for commercial ZnO (black) and ZnO synthesized by bipolar electrochemistry.

**Table 1 materials-12-00535-t001:** The crystalline size (D) calculated from Equation (11); X-ray photoelectron spectroscopy (XPS) chemical shifts of O 1s and Zn 2p signals for commercial ZnO and synthesized by the bipolar electrochemical method. P1 and P2 correspond to resolved peaks of O^2−^ ions in the normal and oxygen-deficient lattice, respectively.

ZnO	D (nm)	O 1s (eV)	P1 (eV)	P2 (eV)	Zn 2p_1/2_ (eV)	Zn 2p_3/2_ (eV)
Com	44	529.2	529.0	531.0	1042.2	1019.2
3He	34	532.1	529.5	532.1	1042.7	1019.7
3Hf	24	530.2	529.0	530.4	1042.8	1019.8
2H	29	530.9	529.5	531.1	1042.7	1019.7

**Table 2 materials-12-00535-t002:** An interfacial charge-transfer resistances (*R_ct_*), charge transport resistances (*R_tr_*), electron diffusion lengths (*L_n_*), and electron lifetimes (*τ**_n_*) are derived from electrochemical characterization of the photoanodes under illumination. The same parameters under dark are demonstrated in Appendix A in supporting information.

ZnO	L (µM)	*R_ct_* (Ω)	*R_tr_* (Ω)	√*R_ct_*/*R_tr_*	*L_n_* (µM × 10^2^)	*τ_n_* (s)
Com	65.5	4558.5	245.0	4.2	2.7	0.27
2H	84.3	994.6	112.2	2.9	2.4	0.21
3He	54.4	4104.4	82.5	7.0	3.8	0.25
3Hf	80.2	1947.7	130.8	3.8	3.0	0.35

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
