# Peer review of "ZnO Synthesized Using Bipolar Electrochemistry: Structure and Activity"

_materials, 2019, doi:10.3390/ma12030535_

Reviewer 1 Report

I found the paper well organized and interesting.

It should be published as it stands.

Author Response

Thank you for your feedback

Reviewer 2 Report

This study describes nanostructured ZnO synthesis. The authors wrote "an eco friendly method" however they used ZnO at the nanoscale.

What about the reproducibility of the synthesis, the ageing and lifetime of the nanostructured ZnO?

Author Response

This study describes nanostructured ZnO synthesis. The authors wrote "an eco friendly method" however they used ZnO at the nanoscale.

Answer:

Very true – nanomaterials are classified as hazards, however they outperform their macroscopic counterpart in many applications, such as photoanodes and photocatalysis.

We replace term “eco friendly” with “reagent-free” in revised version of manuscript (line 392).

What about the reproducibility of the synthesis, the ageing and lifetime of the nanostructured ZnO? 

Answer:

Figure S8 in supporting information demonstrates photos of ZnO suspension prepared from commercial and an in-house made oxide. These solutions were stored in ambient conditions for months and frequently used to prepare new set of electrochemical tests and many materials characterizations. There was not visible change in a color (it would be change from off-white to white) of oxygen-deficient ZnO when exposed to air. This could be easily captured when prepared many photoanode coatings on the FTO plate- but we did not observe any. Also, ZnO with oxygen deficiency has been know to change the oxygen concentration upon exposure to air at temperature higher than 300 oC (see relevant references below), which was never applied in our study. This was also the reason why the anode annealing was done at 250 oC in this work – to prevent from the change in oxygen stoichiometry.

Hong-Li Guo et al in their article: “Oxygen deficient ZnO1−x nanosheets with high visible light photocatalytic activity” https://pubs.rsc.org/en/content/articlehtml/2015/nr/c5nr00271k

“…the oxygen defect-free ZnO sample obtained from the calcination of ZnO1−x at 500 °C in air shows a white color and only responds to ultraviolet light”

Xiaoming Li et al in their article: “Controlling oxygen vacancies and properties of ZnO” https://doi.org/10.1016/j.cap.2014.01.007

“The rising below 300 oC in air is owing to the increase of VO concentration due to the change of chemical equilibrium of decomposition reaction under such special local environment. When the samples were annealed at higher temperature, VO concentration decreased because the outside oxygen defuses into lattice and compensate the vacancy defects.

Reviewer 3 Report

The manuscript entitled "Oxygen-deficient ZnO synthesized using bipolar electrochemistry: structure and activity" reports the synthesis of oxygen defective ZnO by bipolar electrochemistry route and their properties. The objective of the study is good. The approach and obtained results are promising. However, the work lacks in explaining the mechanism behind the observed properties of the materials and the following comments may be useful to improve the manuscript. Therefore, I recommend for a major revision of the manuscript.

1. Add "zinc oxide" in the keywords list

2. Can authors describe the mechanism behind the observed morphology of the synthesized ZnO (Fig. 2a-d)

3. Provide the JCPDS line and plane value of the XRD peaks (Fig. 3)

4. In Fig. 3, the Y axis parameter should be "Intensity (a.u.)" instead of "A.U."

5. It would be appealing if authors could quantify the oxygen defects in the synthesized ZnO structures

6. The red-shift essentially represents the decreasing of band gap energy. Authors should check the discussion on the optical properties

7. The increased band gap energy implies the blue shift and it responds more positively under visible light irradiation as per the data presented in Fig. 6A. It looks implausible.

8. Defects induced optical studies can also be studied using PL. Please provide the PL spectra of the samples

9. In conclusion section, it is mentioned as "The structural and morphological differences influenced the photo-catalytic characteristic"; but photocatalytic data/properties reported

10. Provide the photographic images of the synthesized ZnO materials

Author Response

1. Add "zinc oxide" in the keywords list

Is added in revised manuscript

2. Can authors describe the mechanism behind the observed morphology of the synthesized ZnO (Fig. 2a-d)

Due to the high volume of the paper we excluded detailed discussion on the crystal growth, we assumed it follows the same pathways as in the pioneering work published in 2016 by Abdulaziz et al. (our reference 5) on the bipolar synthesis of ZnO (they obtained star shape micron-sized structures – similar to ours after prolonged exposure of oxide to water). Also, the crystal growth in terms of particle size and shape is similar to structures generated by conventional chemical synthesis methods of ZnO. In this respect, if synthesized in water, the particle size and shape is governed by the rate of dehydration of Zn hydroxyl complexes (according to reaction 2-11 in this work). According to mechanism prosed by others, the rate of dehydration is the main factor affecting the structure of ZnO, as per study cited below. The clarification that dehydration (time that Zn2+ complexes are kept in water) is the most important factor affecting the structure of oxide (not electric field and the time of synthesis upon polarization) appears in revised version of the manuscript in the section related to Fig.2a-d with a new Reference [9b]; lines 170 – 175 refer to:

The structure evolution from the spherical nano-beads to the star-shape larger structures was demonstrated by another group [9b]. The proposed mechanism indicates that upon the prolonged exposure to water the structure of ZnO changes and an initial amount of water affects the nucleation process of ZnO significantly. As observed, an extended contact with water can impede the [0001] growth and accelerate the [1100] growth. This can be controlled by changing water content via mixing it with methanol [9b]. In this way, the shape and size of ZnO can be tailored by adjusting the volume ratio.

For example, significant control on the particle shape and size during the dehydration (drying) of zinc hydroxyl complexes is discussed by Lu et al in their article: “Water Amount Dependence on Morphologies and Properties of ZnO nanostructures in Double-solvent System”

This is also-well discussed by other groups:

The same structures as demonstrated in our work (nano-sized beads and micron size stars formed upon exposed to water) were shown in “The Effect of Solvents, Acetone, Water, and Ethanol, on the Morphological and Optical Properties of ZnO Nanoparticles Prepared by Microwave”

3. Provide the JCPDS line and plane value of the XRD peaks (Fig. 3)

JCPDS reference is added to Fig. 3 in revised version and appear in both new Fig. 3 (black patterns at the bottom of graphs) and in line 219-220 in revised manuscript.

4. In Fig. 3, the Y axis parameter should be "Intensity (a.u.)" instead of "A.U."

New Fig. 3 has corrected label for Y axis – thank you.

5. It would be appealing if authors could quantify the oxygen defects in the synthesized ZnO structures

The only method used in this work for identification of oxygen vacancy is XPS. We can estimate their concertation from the peak ratio P2:P1 in Fig. 4. However, since XPS is the surface technique this quantification should refer to surface concertation only. We stated that oxygen-deficient samples are stable under air, however using XPS for quantitative analysis should be confirmed by other techniques e.g. Raman, EPR or magnetic studies. Due to the large volume of this manuscript, the detailed stoichiometry estimation will be a subject of separate work.

6. The red-shift essentially represents the decreasing of band gap energy. Authors should check the discussion on the optical properties

7. The increased band gap energy implies the blue shift and it responds more positively under visible light irradiation as per the data presented in Fig. 6A. It looks implausible.

Regarding both remarks (#7 and 8): yes, we were wrong with an interpretation of optical results and the correction appears in revised manuscript. We consider that two effects: (1) the particle size, and (2) the oxygen vacancies both influence the observed shifts in the bandgap energy and on the absorption edge. Thus, an interpretation of the optical data presented in Figure 5 accounts for an interplay between structural (oxygen deficiency) and morphological (particle size) effects.

The following correction is applied; lines 262-267:

For the synthesized oxide, the absorption edge is shifted to a slightly shorter wavelength (blue-shift; Figure 5B). The presence of blue-shifting can also indicate on defects in the ZnO lattice, resulting in the formation of shallow levels in the bandgap in this material [14,15]. This is consistent with an increase in bandgap energy for oxides synthesized in this work (Figure 5A). These defects cause the off-white color of the synthesized ZnO (Figure S8). Specifically, yellow ZnO has been related to oxygen deficiencies in the ZnO lattice, resulting in a non-stoichiometric phase Zn1+xO [16].

8. Defects induced optical studies can also be studied using PL. Please provide the PL spectra of the samples

The photoluminescence technique, combined with EPR, magnetic and Raman will be a subject of separate work that will focus only on a structure resolving. Due to large volume of this manuscript relate to photo-electrochemistry we also change the title of manuscript to: “ZnO synthesized by bipolar electrochemistry: structure and activity”

9. In conclusion section, it is mentioned as "The structural and morphological differences influenced the photo-catalytic characteristic"; but photocatalytic data/properties reported

We have changed this statement to (line 391 in revised manuscript):

“The structural and morphological differences influenced the photo-current characteristics”

10. Provide the photographic images of the synthesized ZnO materials

It appears in Figure S8 in supporting information.

Round  2

Reviewer 3 Report

Authors have revised the manuscript satisfactorily and it can be accepted for the publication.